# TESTING RELATIVE FAIRNESS IN HUMAN DECISIONS WITH MACHINE LEARNING

## ABSTRACT

Fairness in decision-making has been a long-standing issue in our society. Compared to algorithmic fairness, fairness in human decisions is even more important since there are processes where humans make the final decisions and that machine learning models inherit bias from the human decisions they were trained on. However, the standard for fairness in human decisions are highly subjective and contextual. This leads to the difficulty for testing "absolute" fairness in human decisions. To bypass this issue, this work aims to test relative fairness in human decisions. That is, instead of defining what are "absolute" fair decisions, we check the relative fairness of one decision set against another. An example outcome can be: Decision Set A favors female over male more than Decision Set B. Such relative fairness has the following benefits: (1) it avoids the ambiguous and contradictory definition of "absolute" fair decisions; (2) it reveals the relative preference and bias between different human decisions; (3) if a reference set of decisions is provided, relative fairness of other decision sets against this reference set can reflect whether those decision sets are fair by the standard of that reference set. We define the relative fairness with statistical tests (null hypothesis and effect size tests) of the decision differences across each sensitive group. Furthermore, we show that a machine learning model trained on the human decisions can inherit the bias/preference and therefore can be utilized to estimate the relative fairness between two decision sets made on different data.

## 1 INTRODUCTION

Recently, much research has been focusing on mitigating the bias and discrimination in machine learning algorithms. This is because machine learning algorithms are increasingly being used to make decisions that affect people's lives and sometimes the learned models behave in a biased manner that gives undue advantages to a specific group of people (where those groups are determined by sex, race, etc.). Such biased decisions can have serious consequences with machine learning algorithms being used in deciding whether a patient gets released from hospital Kharpal (2018), which loan applications are approved Olson (2011), which citizens get bail or sentenced to jail Angwin et al. (2016), who get admitted/hired by universities/companies Dastin (2018).

With the emerging research effort on detecting and mitigating bias in machine learning algorithms, bias and discrimination can be reduced for processes where the machine learning models and software directly make the final decisions, such as loan approvals on Wonga.com Olson (2011). However, there are processes where human makes the final decisions. For example, the human resource person makes the final decision on which applicant gets an interview while a machine learning software assists the process by ranking the applicants with its predictions Dastin (2018). Reducing bias in machine learning models alone does not make these processes fair as long as the responsible human continues to make biased final decisions. Furthermore, what is considered to be "absolute fair" for human decisions are highly subjective and contextual Abu-Elyounes (2020). For example, should universities admit students without considering gender/race or should they admit same percentage of students from each gender/race group? These are often contradictory standards of fair decisions. To name a few common ones, individual fairness Dwork et al. (2012) requires similar individuals (from different sensitive groups) to be treated similarly, demographic parity Dwork et al. (2012) requires the acceptance rates to be the same across different sensitive groups, other group fairness notions such as equalized odds Hardt et al. (2016) require a ground truth decision set to evaluate. In many

scenarios, these "absolute" fairness notions are contradictory and impossible to be satisfied at the same time Friedler et al. (2021).

To bypass this issue, this work aims to test relative fairness in human decisions. That is, instead of defining what are "absolute" fair decisions, we check the relative fairness of one decision set against another. To do this, we focus on the differences between the two decisions sets— the relative fairness is defined as the statistical parity of the differences between two sets of decisions over a certain sensitive attribute. Detailed definition will be provided in Section 3.1. The intuition behind this definition is that, the existence of such difference parity indicates that one decision set is constantly overrating one sensitive group than another when compared to the other decision set. This definition of relative fairness has the following benefits: (1) it avoids the ambiguous and contradictory definition of what absolutely fair decisions are; (2) it reveals the relative preference and bias between different human decisions; (3) if a reference set of decisions is provided, relative fairness of other decision sets against this reference set can reflect whether those decisions sets are fair by the standard of that reference set. Note that, this work does not evaluate relative fairness between decisions made for different tasks or in different contexts. In the rest of the paper, when we say decisions made on different data, we mean different data items from the same dataset (for the same task).

One limitation of the relative fairness defined in this paper is that it requires the two set of decisions to be made on the same data. Even for the same task, such overlapping decisions are not always available— e.g. there might be multiple HRs screening for the same job application, but one application will only be screened by one HR. To overcome this limitation, we propose a testing framework of the relative fairness between Human $Y_0$'s decisions on data $X_0$ and Human $Y_1$'s decisions on data $X_1$ by (1) fitting a machine learning model $f(X)$ with $X_0$ and $Y_0$; (2) testing the relative fairness of the model's predictions on data $X_1$— $f(X_1)$ against $Y_1$. Both theoretically and empirically, we show that, the relative fairness of the model's predictions $f(X_1)$ against $Y_1$ could be utilized to correctly estimate the relative fairness between $Y_0$ and $Y_1$.

## 1.1 MOTIVATION

In this section, we demonstrate the potential application of the proposed relative fairness with the following example scenarios.

**Scenario 1: Decisions from different humans are made on the same data.** For example, multiple human resource (HR) persons screen the same set of applicants. In this scenario, relative fairness defined in Section 3.1 between pairs of HRs can be tested to learn about the relative biases/preferences. If the HRs have reached consensus decisions, the relative bias between each HR and the consensus decisions can also reflect the initial biases/preferences that HR need to reduce in the future.

**Scenario 2: Decisions from different humans are made on different data (for the same task).** For example, screening decisions for the same job application from two consecutive years. The relative fairness between the decisions of the second year and those of the first year can be tested by our proposed algorithm in Section 3.3 to check whether they are consistent. Furthermore, this relative fairness can also be utilized to achieve a specific fairness goal, e.g. increasing the female employee rate over the last year.

## 1.2 CONTRIBUTIONS

The contributions of this work include:

- The definition of relative fairness that can be applied without the need of ground truth or the standard of what "absolute" fairness is.
- Metrics measuring the violation of the proposed relative fairness.
- Two machine learning-based algorithms to estimate relative fairness between decisions made on different data for the same task.
- The theoretical analyses of how the proposed relative fairness estimation algorithm works.
- The empirical results also demonstrate the consistency of the relative fairness metrics and the effectiveness of estimating the relative fairness of human decisions with the proposed algorithm.

- The code and data used in this work are publicly available[1].

## 2 BACKGROUND AND RELATED WORK

Research on human decision fairness is difficult. This is because ground truth for human decisions is impossible or prohibitively expensive to obtain, and so bias is hard to directly measure. For example, Sap et al. Sap et al. (2019) study several datasets of social media posts annotated for the presence of hate speech. They showed that when the posts are written in the African American English (AAE) dialect or are authored by users who self identify as Black the posts are more likely to be labeled (both by human annotators and the models learned from them) as hate speech than if they are not written in AAE or are authored by users who self identify as White. Two different reasons could lead to this finding— (1) the human annotators are biased towards AAE dialect or black post authors; or (2) posts authored by users who self identify as Black or written in the AAE dialect tend to be more offensive. If it is caused by the first reason, we want to fix such bias caused by annotators. If it is caused by the second reason, the annotations should be considered correct. However, without the ground truth, there is no way to know which reason leads to this finding.

Attempts have been made to estimate the ground truth by acquiring multiple annotations on the same data. Several studies have shown that, for binary labeling, 3–10 annotators per item is sufficient to obtain reliable labels (evaluated using inter-annotator agreement scores Artstein (2017) such as Cohen's kappa and Krippendorff's alpha). Dawid and Skene Dawid & Skene (1979) used the EM algorithm to iteratively estimate the ground ground labels, along with the (two sided) error rate of each annotator, for binary labeling problems. This model has been later extended by other researchers for other scenarios Kairam & Heer (2016); Carpenter (2008); Raykar et al. (2010); Weld et al. (2011); Ipeirotis et al. (2010); Pasternack & Roth (2010); Felt et al. (2014); Hovy et al. (2013). Liu and others have taken the more radical approach of treating the ground truth of each label not as a single value, but as a distribution over the answers that a population of (mostly hidden) annotators would provide, where the actual labels obtained are merely an observed sample of this hidden population's responses Liu et al. (2019); Weerasooriya et al. (2020; 2021). However, these approaches require multiple annotations on the same data, which is expensive and does not scale up well.

Given the difficulty of obtaining the ground truth for human decisions. We would instead analyze the relative bias between different annotators— the difference in preference/bias across annotators. As an example, Price and Wolfers Price & Wolfers (2010) showed that, white NBA referees tend to award more extra fouls towards black players than black NBA referees. In this case, we can say that those white NBA referees has a relative bias of awarding more fouls towards black players than white players when compared to black NBA referees. On the other hand, another study by Welch et al. Welch et al. (1988) showed that black and white judges weighted case and offender information in similar ways when making punishment decisions, although black judges were more likely to sentence both black and white offenders to prison. In this case, the black judges do not have a relative bias towards black or white offenders when compared to white judges.

Previously, there is no clear definition of relative fairness or relative bias. And these previous studies analyzed data without strict controls. For example, Price and Wolfers Price & Wolfers (2010) analyzed the data by regressing the number of fouls called per 48 minutes for each player-game observation in which the referee participated, against an indicator variable for whether the offending player is black. The data for each referee come from different games played by different players. The conclusion can be misled by coincidences such as white players happened to commit more fouls in games with black referees. To avoid such ambiguity, we define relative fairness as difference parity— the differences between the two sets of decisions made on the same data are statistically the same across different sensitive groups. This relative fairness definition has no assumption on the task and thus is more general than the existing "absolute" fairness definitions. For example, individual fairness Dwork et al. (2012) requires similar individuals (from different sensitive groups) to be treated similarly, demographic parity Dwork et al. (2012) requires the acceptance rates to be the same across different sensitive groups, other group fairness notions such as equalized odds Hardt et al. (2016) require the ground truth decisions to evaluate. These "absolute" fairness notions are often contradictory and impossible to be satisfied at the same time Friedler et al. (2021) but relative fairness can be evaluated regardless these requirements.

---

[1] https://anonymous.4open.science/r/ContextualFairnessTesting-2B8C

## 3 METHODOLOGY

### 3.1 RELATIVE FAIRNESS

We define the relative fairness as difference parity— the differences between the two sets of decisions are statistically the same across different sensitive groups. Without loss of generality, we assume a binary sensitive attribute in the following definition.

**Definition 3.1.** *Relative fairness. Given a set of data $X \in \mathbb{R}^d$ with binary sensitive attribute $A(X) \in \{0, 1\}$, and two sets of decisions on the data $Y_0(X)$, $Y_1(X) \in \mathbb{R}$, the two decision sets are considered as relatively fair to each other if and only if the difference of the decisions on each sensitive group share the same mean value:*

$$\mu(Y_\Delta(A = 0)) = \mu(Y_\Delta(A = 1)) = \mu$$

*where*

$$Y_\Delta(A = a) = \{Y_0(x) - Y_1(x) | A(x) = a, x \in X\},$$

*and $\mu(Y_\Delta)$ is the mean of the underlying distribution of $Y_\Delta$.*

Following the above definition, two decision sets are considered as relatively fair if they are not overestimating or underestimating any sensitive group compared to each other. Since every decision is made independently, the differences of the decisions $Y_\Delta(A = a)$ are independent and identically distributed (i.i.d.). Based on central limit theorem and the law of large numbers, the sampled mean of $Y_\Delta(A = a)$ follows normal distribution in large samples:

$$\frac{\overline{Y}_\Delta(A = a) - \mu(Y_\Delta(A = a))}{s(\overline{Y}_\Delta(A = a))} \xrightarrow{d} \mathcal{N}(0, 1). \tag{1}$$

where $s(\overline{Y}_\Delta(A = a)) = \sqrt{\frac{s^2(Y_\Delta(A=a))}{|A(X)=a|}}$ is the sampled standard deviation of $\overline{Y}_\Delta(A = a)$ and $s(Y_\Delta(A = a))$ is the sampled standard deviation of $Y_\Delta(A = a)$.

Given equation 1, we define two relative bias metrics to measure 1) the probability of the difference between $\overline{Y}_\Delta(A = 0)$ and $\overline{Y}_\Delta(A = 1)$ arise from random chance with null hypothesis testing Anderson et al. (2000); and 2) the strength of the difference with effect size testing Chow (1988). Welch's t-test Welch (1947) and Cohen's d Cohen (2013) are applied to test the null hypothesis and effect size.

### 3.2 RELATIVE BIAS METRICS

**Definition 3.2.** *Null hypothesis testing for relative bias. Given a set of data $X$ with sensitive attribute $A(X) \in \{0, 1\}$, and two sets of decisions on the data $Y_0(X)$, $Y_1(X)$, the relative bias t (RBT) score of $Y_0$ over $Y_1$ on $A(X) \in \{0, 1\}$ is calculated as equation 2.*

$$RBT(Y_0, Y_1, A) = \frac{\overline{Y}_\Delta(A = 1) - \overline{Y}_\Delta(A = 0)}{\sqrt{s^2(\overline{Y}_\Delta(A = 1)) + s^2(\overline{Y}_\Delta(A = 0))}}$$

$$DoF(Y_0, Y_1, A) = \frac{(s^2(\overline{Y}_\Delta(A = 1)) + s^2(\overline{Y}_\Delta(A = 0)))^2}{\frac{(s^2(\overline{Y}_\Delta(A=1)))^2}{|A(X)=1|-1} + \frac{(s^2(\overline{Y}_\Delta(A=0)))^2}{|A(X)=0|-1}}. \tag{2}$$

**Definition 3.3.** *Effect size for relative bias. Given a set of data $X$ with sensitive attribute $A(X) \in \{0, 1\}$, and two sets of decisions on the data $Y_0(X)$, $Y_1(X)$, the relative bias d (RBD) score of $Y_0$ over $Y_1$ on $A(X) \in \{0, 1\}$ is calculated as equation 3.*

$$RBD(Y_0, Y_1, A) = \frac{\overline{Y}_\Delta(A = 1) - \overline{Y}_\Delta(A = 0)}{s} \tag{3}$$

*where $s = \sqrt{\frac{(|A(X)=1|-1)s^2(Y_\Delta(A=1)) + (|A(X)=0|-1)s^2(Y_\Delta(A=0))}{|A(X)=1| + |A(X)=0| - 2}}$ is the pooled standard deviation.*

**Relative bias:** Utilizing the two metrics, $Y_0$ is relatively biased towards $A = 1$ compared to $Y_1$ if the null hypothesis is rejected at more than 95% confidence—one tailed $p \leq 0.05$ given the t value $RBT(Y_0, Y_1, A)$ and degree of freedom $DoF(Y_0, Y_1, A)$, vice versa. The magnitude of the relative bias will be determined by the $RBD$ value following the same magnitude descriptor as Cohen's d in Table 1.

Table 1: Effect sizes Sawilowsky (2009).

| Effect Size | d | | Effect Size | d | |
|---|---|---|---|---|---|
| Very Small | 0.01 | ‖ | Large | 0.8 | |
| Small | 0.2 | ‖ | Very Large | 1.2 | |
| Medium | 0.5 | ‖ | Huge | 2.0 | |

Note that, in the binary classification setting $Y_0, Y_1 \in \{0, 1\}$, this definition of relative bias is related to demographic parity since

$$\overline{Y}_\Delta(A = 1) - \overline{Y}_\Delta(A = 0) = (\overline{Y_0}(A = 1) - \overline{Y_1}(A = 1)) - (\overline{Y_0}(A = 0) - \overline{Y_1}(A = 0))$$
$$= DP(Y_0) - DP(Y_1).$$

However, it is still different from $DP(Y_0) - DP(Y_1)$ because relative bias also takes into consideration the variance of the differences which is unavailable from the statistics of $Y_0$ and $Y_1$.

### 3.3 ESTIMATING RELATIVE FAIRNESS BETWEEN DECISIONS MADE ON DIFFERENT DATA

**Problem statement:** Given two non-overlapping data $X_0$ and $X_1$ drawn from the same distribution, and two decision sets $Y_0(X_0)$ and $Y_1(X_1)$ made by different humans, test the relative fairness/bias between the two decision sets over a certain sensitive attribute $A$.

#### 3.3.1 BIASED BRIDGE

The first approach, biased bridge, utilizes a machine learning model $f(x)$ to bridge the two decision sets made on different data $X_0$ and $X_1$. Being trained on $(X_0, Y_0(X_0))$, $f(x)$ could make predictions for both $f(X_0)$ and $f(X_1)$. Given that the errors of the model's predictions are also i.i.d., their sampled means should follow normal distribution in large samples:

$$\frac{\overline{E}_0(A = a) - \mu(E_0(A = a))}{s(\overline{E}_0(A = a))} \xrightarrow{d} \mathcal{N}(0, 1) \tag{4}$$

$$\frac{\overline{E}_1(A = a) - \mu(E_1(A = a))}{s(\overline{E}_1(A = a))} \xrightarrow{d} \mathcal{N}(0, 1) \tag{5}$$

where $E_0(A = a) = \{f(x) - Y_0(x)|A(x) = a\}$ and $E_1(A = a) = \{f(x) - Y_1(x)|A(x) = a\}$ are the errors of $f(x)$ compared to $Y_0$ and $Y_1$. Substract $\overline{E}_0(A = a)$ from $\overline{E}_1(A = a)$ we have

$$\frac{\overline{E}_1(A = a) - \overline{E}_0(A = a)}{\sqrt{s^2(\overline{E}_1(A = a)) + s^2(\overline{E}_0(A = a))}} \xrightarrow{d} \mathcal{N}(0, 1). \tag{6}$$

Given that

$$Y_\Delta(A = a) = \{Y_0(x) - Y_1(x)|A(x) = a\} = E_1(A = a) - E_0(A = a),$$

we can estimate:

$$\overline{Y}_\Delta(A = a) \hat{=} \overline{E}_1(A(x) = a, x \in X_1) - \overline{E}_0(A(x) = a, x \in X_0)$$
$$s(\overline{Y}_\Delta(A = a)) \hat{=} \sqrt{s^2(\overline{E}_1(A(x) = a, x \in X_1)) + s^2(\overline{E}_0(A(x) = a, x \in X_0))}. \tag{7}$$

As shown in Algorithm 1, the relative bias metrics can then be calculated as equation 2 and equation 3.

**Algorithm 1:** Biased Bridge.

**Input** : $(X_0, A(X_0), Y_0(X_0))$.
    $(X_1, A(X_1), Y_1(X_1))$.
    $f(x)$, a predictor.
**Output :** Relative bias of $Y_0$ over $Y_1$.

1 f(x).fit($X_0, Y_0(X_0)$)
2 Estimate $\overline{Y}_\Delta(A = a)$ as equation 7
3 Estimate $s(\overline{Y}_\Delta(A = a))$ as equation 7
4 **return** RBT($Y_0, Y_1$, A), RBD($Y_0, Y_1$, A)

**Algorithm 2:** Unbiased Bridge.

**Input** : $(X_0, A(X_0), Y_0(X_0))$.
    $(X_1, A(X_1), Y_1(X_1))$.
    $f(x)$, a predictor.
**Output :** Relative bias of $Y_0$ over $Y_1$.

1 f(x).fit($X_0, Y_0(X_0)$)
2 rbt = RBT(f($X_1$), $Y_1(X_1)$, A)
3 rbd = RBD(f($X_1$), $Y_1(X_1)$, A)
4 **return** rbt, rbd

### 3.3.2 UNBIASED BRIDGE

The second approach, unbiased bridge, utilizes the same machine learning model $f(x)$ trained on $(X_0, Y_0(X_0))$. In this approach, we simply assume that $f(x)$ has inherited every human bias in $Y_0(X_0)$. As a result, the predictions of $f(X_1)$ will be treated as $Y_0(X_1)$ and will be used to compare against $Y_1(X_1)$ for relative fairness. As shown in Algorithm 2, $RBT(Y_0, Y_1, A)$ and $RBD(Y_0, Y_1, A)$ are estimated as $RBT(f(X_1), Y_1(X_1), A)$ and $RBD(f(X_1), Y_1(X_1), A)$.

## 4 EXPERIMENT DESIGN

Table 2: Description of the datasets used in the experiments.

| Dataset | #Rows | Sensitive Attributes $A$ | | Class Labels $Y$ | |
|---|---|---|---|---|---|
| | | $A = 0$ | $A = 1$ | | |
| Adult Census Income Dua & Graff (2017) | 48,842 | Sex-Female Race-Nonwhite | Sex-Male Race-White | $Y = 1$ Income > 50K | $Y = 0$ Income $\leq$ 50K |
| SCUT-FBP5500 Liang et al. (2018) | 5,500 | Sex-Female Race-Asian | Sex-Male Race-Caucasian | Beauty Rating $Y \in \{1, 2, 3, 4, 5\}$ | |

In this section, we design experiments on two datasets[2] shown in Table 2 to explore:

- **RQ1:** Does the proposed relative fairness metrics consistently reflect the relative fairness between two sets of decisions made on the same data?

- **RQ2:** Do the proposed frameworks, biased bridge and unbiased bridge, correctly estimate the relative fairness between the decisions made on different data?

### 4.1 EXPERIMENT WITH SYNTHETIC RELATIVE BIAS

The Adult Census Income data only has one set of decisions $Y_0$ which comes from ground truth. The data is randomly split into 70% for training ($X_0$) and 30% for testing ($X_1$). After that, we manually inject bias into the labels as $Y_1$ where

$$P(Y_1(A = a) = 1) = Y_0(A = a) + \mathcal{N}(\lambda(A) \cdot s(Y_0) \cdot z(a), (\lambda(A) \cdot s(Y_0))^2)$$

where $s(Y_0)$ is the standard deviation of $Y_0$, $\lambda(A)$ is a parameter controlling the mean of the injected bias to different sensitive groups, and

$$z(a) = \frac{a - \overline{A}}{s(A)}.$$

A logistic regression model $f(x)$ is trained on the biased training data $Y_1(X_0)$. The training time is less than 1 second on a single desktop. The learned model is then being utilized in unbiased bridge and biased bridge to estimate the relative fairness between $Y_1$ and $Y_0$.

---

[2]Experimental results with injected bias on 7 more datasets are available at `https://anonymous.4open.science/r/ContextualFairnessTesting-2B8C`.

### 4.2 Experiment with Real Human Decision Bias

The SCUT-FBP5500 dataset has 5,500 face images and their beauty ratings from 60 different human raters. In this experiment, we utilized the ratings from the first three humans (P1, P2, and P3) and the average ratings of the 60 humans (Average) as four different decision sets. These ratings range from 1 (least beautiful to the rater) to 5 (most beautiful to the rater), therefore it is a regression problem. Note that all the analyses and definitions in Section 3 apply to both regression and binary classification problems. The image set is randomly split into 80% for training ($X_0$) and 20% for testing ($X_1$). A VGG-16 model Simonyan & Zisserman (2014) with pre-trained weights on the ImageNet data is transferred to predict the beauty ratings with the output layer being replaced as a dense layer of size 256 and a one node output layer. Each time, one VGG-16 model is trained to learn from the a specific set of decisions on the training set $X_0$. Then its predictions on the test set $X_1$ are compared against another set of decisions on the test set $X_1$ to estimate relative fairness between these two sets of decisions. These models are trained to minimize binary cross entropy loss for 1,000 epochs with batch size = 10. The training time of adapting the VGG-16 model to one decision set was around 2.2 hours on four NVIDIA A100 Tensor Core GPUs.

## 5 Experimental Results

Table 3 shows the experimental results on the Adult Census Income dataset with injected bias. Table 4 shows the experimental results on the SCUT-FBP5500 dataset with four decision sets (P1, P2, P3, and Average). Each experiment is conducted only once due to the high computation cost and to better simulate what will happen in practice. In these tables, "GT Train" represents the ground truth relative fairness metrics calculated with $Y_0(X_0)$ and $Y_1(X_0)$ on the training data; "GT Test" represents the ground truth relative fairness metrics calculated with $Y_0(X_1)$ and $Y_1(X_1)$ on the test data. "Biased Bridge" and "Unbiased Bridge" represents the relative fairness metrics calculated by the proposed frameworks in Algorithm 1 and 2. Every result reports the p value of the null hypothesis test in round brackets, followed by the effect size testing result. One result is left in white background when no significant relative bias is found (p value > 0.05); is colored in green if $Y_0$ is found to favor $A = 1$ more when compared to $Y_1$ (p value ≤ 0.05 and RBD>0); and is colored in red if $Y_0$ is found to favor $A = 1$ less when compared to $Y_1$ (p value ≤ 0.05 and RBD<0). With results from the two tables, we answer the two research questions from the previous section.

Table 3: Results on the Adult Census Income data. Results are shown with numbers of (p value) and RBD value. Results with p value ≤ 0.05 are colored as green if RBD>0 or red if RBD<0.

| $\lambda(A)$ | GT Train | | GT Test | | Unbiased Bridge | | Biased Bridge | |
|---|---|---|---|---|---|---|---|---|
| | sex | race | sex | race | sex | race | sex | race |
| $\lambda$(sex)=0, $\lambda$(race)=0 | (0.50) 0.00 | (0.50) 0.00 | (0.50) 0.00 | (0.50) 0.00 | (0.45) -0.00 | (0.43) -0.00 | (0.47) -0.00 | (0.45) -0.00 |
| $\lambda$(sex)=0.1, $\lambda$(race)=0 | (0.00) 0.20 | (0.34) -0.01 | (0.00) 0.20 | (0.08) 0.04 | (0.00) 0.12 | (0.04) -0.04 | (0.00) 0.08 | (0.11) -0.03 |
| $\lambda$(sex)=0.2, $\lambda$(race)=0 | (0.00) 0.29 | (0.29) 0.01 | (0.00) 0.28 | (0.38) 0.01 | (0.00) 0.19 | (0.42) 0.01 | (0.00) 0.13 | (0.44) 0.00 |
| $\lambda$(sex)=-0.1, $\lambda$(race)=0 | (0.00) -0.37 | (0.00) -0.10 | (0.00) -0.34 | (0.00) -0.10 | (0.00) -0.19 | (0.00) -0.08 | (0.00) -0.13 | (0.02) -0.05 |
| $\lambda$(sex)=-0.2, $\lambda$(race)=0 | (0.00) -0.52 | (0.00) -0.13 | (0.00) -0.52 | (0.00) -0.10 | (0.00) -0.36 | (0.00) -0.09 | (0.00) -0.24 | (0.01) -0.06 |
| $\lambda$(sex)=0, $\lambda$(race)=0.1 | (0.01) -0.04 | (0.00) 0.24 | (0.03) -0.04 | (0.00) 0.25 | (0.24) -0.01 | (0.00) 0.11 | (0.32) -0.01 | (0.00) 0.07 |
| $\lambda$(sex)=0, $\lambda$(race)=0.2 | (0.03) -0.03 | (0.00) 0.33 | (0.00) -0.06 | (0.00) 0.32 | (0.21) -0.02 | (0.00) 0.21 | (0.30) -0.01 | (0.00) 0.14 |
| $\lambda$(sex)=0, $\lambda$(race)=-0.1 | (0.00) -0.11 | (0.00) -0.59 | (0.00) -0.07 | (0.00) -0.63 | (0.01) -0.05 | (0.00) -0.30 | (0.05) -0.03 | (0.00) -0.21 |
| $\lambda$(sex)=0, $\lambda$(race)=-0.2 | (0.00) -0.14 | (0.00) -0.75 | (0.00) -0.14 | (0.00) -0.80 | (0.00) -0.11 | (0.00) -0.49 | (0.00) -0.07 | (0.00) -0.33 |
| $\lambda$(sex)=0.2, $\lambda$(race)=0.2 | (0.00) 0.27 | (0.00) 0.34 | (0.00) 0.29 | (0.00) 0.32 | (0.00) 0.19 | (0.00) 0.23 | (0.00) 0.13 | (0.00) 0.16 |
| $\lambda$(sex)=-0.2, $\lambda$(race)=-0.2 | (0.00) -0.49 | (0.00) -0.68 | (0.00) -0.51 | (0.00) -0.74 | (0.00) -0.41 | (0.00) -0.57 | (0.00) -0.28 | (0.00) -0.38 |
| $\lambda$(sex)=0.2, $\lambda$(race)=-0.2 | (0.00) 0.19 | (0.00) -0.58 | (0.00) 0.17 | (0.00) -0.64 | (0.00) 0.15 | (0.00) -0.40 | (0.00) 0.10 | (0.00) -0.27 |
| $\lambda$(sex)=-0.2, $\lambda$(race)=0.2 | (0.00) -0.45 | (0.00) 0.29 | (0.00) -0.46 | (0.00) 0.28 | (0.00) -0.36 | (0.00) 0.23 | (0.00) -0.24 | (0.00) 0.15 |

**RQ1: Does the proposed relative fairness metrics consistently reflect the relative fairness between two sets of decisions made on the same data?**

As stated in Section 3.1, a relative bias is considered to be statistically significant when the $p \leq 0.05$ given its null hypothesis test result. In Table 3, all of the 26 cells show consistent results between "GT Train" and "GT Test". This suggests that given the same form of injected bias, the relative biases between the ground truth decisions and the synthetic decisions tested are always consistent when tested on different data.

In Table 4, we observed that in most cases, the relative bias metrics between each pair of decision makers (e.g. P1 vs. P2) are consistent on different data $X_0$ and $X_1$. For example, according to GT

Table 4: Results on the SCUT-FBP5500 data. Results are shown with numbers of (p value) and RBD value. Results with p value $\leq 0.05$ are colored as green if RBD$>0$ or red if RBD$<0$.

| | | Sex | Race | Sex | Race | Sex | Race | Sex | Race |
|---|---|---|---|---|---|---|---|---|---|
| | | P1 | | P2 | | P3 | | Average | |
| GT Train | P1 | (0.50) 0.00 | (0.50) 0.00 | (0.00) -0.45 | (0.00) -0.25 | (0.00) -0.50 | (0.01) -0.07 | (0.00) -0.48 | (0.00) -0.25 |
| GT Test | | (0.50) 0.00 | (0.50) 0.00 | (0.00) -0.39 | (0.00) -0.19 | (0.00) -0.44 | (0.03) -0.13 | (0.00) -0.43 | (0.00) -0.24 |
| Unbiased Bridge | | (0.04) -0.11 | (0.27) -0.04 | (0.00) -0.58 | (0.00) -0.28 | (0.00) -0.57 | (0.01) -0.18 | (0.00) -0.68 | (0.00) -0.36 |
| Biased Bridge | | (0.10) -0.08 | (0.44) -0.01 | (0.00) -0.52 | (0.00) -0.22 | (0.00) -0.52 | (0.03) -0.14 | (0.00) -0.57 | (0.00) -0.27 |
| GT Train | P2 | (0.00) 0.45 | (0.00) 0.25 | (0.50) 0.00 | (0.50) 0.00 | (0.04) -0.05 | (0.00) 0.17 | (0.00) 0.14 | (0.00) 0.11 |
| GT Test | | (0.00) 0.39 | (0.00) 0.19 | (0.50) 0.00 | (0.50) 0.00 | (0.16) -0.06 | (0.17) 0.06 | (0.07) 0.09 | (0.48) -0.00 |
| Unbiased Bridge | | (0.00) 0.40 | (0.00) 0.22 | (0.11) 0.07 | (0.15) 0.07 | (0.33) -0.03 | (0.09) 0.09 | (0.01) 0.15 | (0.17) 0.06 |
| Biased Bridge | | (0.00) 0.40 | (0.00) 0.22 | (0.12) 0.07 | (0.15) 0.07 | (0.37) -0.02 | (0.08) 0.10 | (0.01) 0.14 | (0.16) 0.07 |
| GT Train | P3 | (0.00) 0.50 | (0.01) 0.07 | (0.04) 0.05 | (0.00) -0.17 | (0.50) 0.00 | (0.50) 0.00 | (0.00) 0.16 | (0.00) -0.16 |
| GT Test | | (0.00) 0.44 | (0.03) 0.13 | (0.16) 0.06 | (0.17) -0.06 | (0.50) 0.00 | (0.50) 0.00 | (0.02) 0.13 | (0.11) -0.08 |
| Unbiased Bridge | | (0.00) 0.45 | (0.02) 0.14 | (0.18) 0.05 | (0.10) -0.08 | (0.36) -0.02 | (0.49) -0.00 | (0.01) 0.14 | (0.06) -0.11 |
| Biased Bridge | | (0.00) 0.43 | (0.01) 0.15 | (0.20) 0.05 | (0.29) -0.04 | (0.40) -0.02 | (0.33) 0.03 | (0.03) 0.12 | (0.25) -0.05 |
| GT Train | Average | (0.00) 0.48 | (0.00) 0.25 | (0.00) -0.14 | (0.00) -0.11 | (0.00) -0.16 | (0.00) 0.16 | (0.50) 0.00 | (0.50) 0.00 |
| GT Test | | (0.00) 0.43 | (0.00) 0.24 | (0.07) -0.09 | (0.48) 0.00 | (0.02) -0.13 | (0.11) 0.08 | (0.50) 0.00 | (0.50) 0.00 |
| Unbiased Bridge | | (0.00) 0.36 | (0.00) 0.23 | (0.01) -0.13 | (0.25) 0.05 | (0.01) -0.14 | (0.09) 0.10 | (0.11) -0.07 | (0.20) 0.06 |
| Biased Bridge | | (0.00) 0.37 | (0.00) 0.23 | (0.05) -0.10 | (0.27) 0.04 | (0.02) -0.12 | (0.10) 0.09 | (0.30) -0.03 | (0.23) 0.05 |

Train (P1, P2), P1 favors Female (Sex=0) over Male (Sex=1) when compared to P2 ($p = 0.00 < 0.05$, $RBD = -0.45$) and also favors Asian (Race=0) over Caucasian (Race=1) when compared to P2 ($p = 0.00 < 0.05$, $RBD = -0.25$). According to GT Test(P1, P2), P1 favors Female (Sex=0) over Male (Sex=1) when compared to P2 ($p = 0.00 < 0.05$, $RBD = -0.39$) and also favors Asian (Race=0) over Caucasian (Race=1) when compared to P2 ($p = 0.00 < 0.05$, $RBD = -0.19$). These two results are consistent. Out of 32 pairwise comparisons of GT Train and GT Test, 22 of them are consistent while 10 are not. Given the randomness in splitting the training and testing data and the fact that all ten inconsistencies do not have opposite relative bias results, overall, the proposed relative fairness metrics consistently reflect the relative fairness in the SCUT-FBP5500 dataset.

> **To RQ1.** The proposed relative fairness metrics consistently reflect the relative fairness between two sets of decisions made on the same data in most cases.

Table 5: Confusion Matrices.

| | | Adult Census Income | | | SCUT-FBP5500 | | |
|---|---|---|---|---|---|---|---|
| | | GT $> 0$ | GT $< 0$ | GT $= 0$ | GT $> 0$ | GT $< 0$ | GT $= 0$ |
| Unbiased Bridge | EST $> 0$ | 8 | 0 | 0 | 8 | 0 | 0 |
| | EST $< 0$ | 0 | 12 | 1 | 0 | 8 | 1 |
| | EST $= 0$ | 0 | 2 | 3 | 0 | 0 | 15 |
| Biased Bridge | EST $> 0$ | 8 | 0 | 0 | 8 | 0 | 0 |
| | EST $< 0$ | 0 | 12 | 0 | 0 | 8 | 0 |
| | EST $= 0$ | 0 | 2 | 4 | 0 | 0 | 16 |

Table 6: Training Performances.

| Data | Train on | MAE | Sex | Race |
|---|---|---|---|---|
| Adult Census Income | $\lambda$(sex)=0, $\lambda$(race)=0 | 0.20 | (0.50) -0.00 | (0.50) -0.00 |
| | $\lambda$(sex)=0.1, $\lambda$(race)=0 | 0.22 | (0.50) -0.00 | (0.50) -0.00 |
| | $\lambda$(sex)=0.2, $\lambda$(race)=0 | 0.25 | (0.49) -0.00 | (0.50) -0.00 |
| | $\lambda$(sex)=-0.1, $\lambda$(race)=0 | 0.24 | (0.50) 0.00 | (0.50) -0.00 |
| | $\lambda$(sex)=-0.2, $\lambda$(race)=0 | 0.28 | (0.50) 0.00 | (0.50) -0.00 |
| | $\lambda$(sex)=0, $\lambda$(race)=0.1 | 0.23 | (0.50) -0.00 | (0.50) -0.00 |
| | $\lambda$(sex)=0, $\lambda$(race)=0.2 | 0.26 | (0.50) -0.00 | (0.50) -0.00 |
| | $\lambda$(sex)=0, $\lambda$(race)=-0.1 | 0.24 | (0.50) -0.00 | (0.50) 0.00 |
| | $\lambda$(sex)=0, $\lambda$(race)=-0.2 | 0.26 | (0.50) -0.00 | (0.50) 0.00 |
| | $\lambda$(sex)=0.2, $\lambda$(race)=0.2 | 0.27 | (0.49) -0.00 | (0.50) -0.00 |
| | $\lambda$(sex)=-0.2, $\lambda$(race)=-0.2 | 0.30 | (0.50) -0.00 | (0.50) 0.00 |
| | $\lambda$(sex)=0.2, $\lambda$(race)=-0.2 | 0.27 | (0.49) -0.00 | (0.50) 0.00 |
| | $\lambda$(sex)=-0.2, $\lambda$(race)=0.2 | 0.29 | (0.50) 0.00 | (0.50) -0.00 |
| SCUT-FBP5500 | P1 | 0.02 | (0.00) -0.21 | (0.00) -0.40 |
| | P2 | 0.04 | (0.17) -0.03 | (0.49) 0.00 |
| | P3 | 0.02 | (0.00) -0.11 | (0.00) -0.20 |
| | Average | 0.01 | (0.00) -0.11 | (0.19) 0.03 |

**RQ2: Do the proposed frameworks, unbiased bridge and biased bridge, correctly estimate the relative fairness between the decisions made on different data?**

Table 5 shows the confusion matrices of unbiased bridge and biased bridge estimations summarized from Table 3 and Table 4. EST>0 means a positive relative bias is detected with $p \leq 0.05$ and RBD>0. EST<0 means a negative relative bias is detected with $p \leq 0.05$ and RBD<0. EST=0 means no significant relative bias is detected ($p > 0.05$). GT>0 means either GT Train or GT Test has confirmed the detected positive relative bias; GT<0 means either GT Train or GT Test has confirmed the detected negative relative bias; GT $= 0$ means either GT Train or GT Test has confirmed that no significant relative bias exits. From Table 5, we observe that biased bridge estimation has $100\%$ precision in detecting relative biases on both datasets. It also has $91\%$ recall on the Adult Census Income data and $100\%$ recall on the SCUT-FBP5500 data. On the other hand, the unbiased bridge estimation has $95\%$ precision and $91\%$ recall on the Adult Census Income data and $94\%$ precision and $100\%$ recall on the SCUT-FBP5500 data. Overall, biased bridge estimation has better accuracy in estimating the relative fairness between the decisions made on different data.

Table 6 shows the training performances of the logistic regression classifier on Adult Census Income data and the VGG-16 regressor on SCUT-FBP5500 data. From this table we can see that, the logistic regression classifier has no relative bias to its training data but has a large training error rate. On the other hand, the VGG-16 regressor has low training error but significant relative bias to its training data. Both high training error or significant relative bias to the training data can cause inaccurate estimations from the unbiased bridge approach. This explains the better performance of biased bridge estimation and the necessity of using it.

> **To RQ2. Overall, both approaches achieved good accuracy in estimating the relative fairness between the decisions made on different data. However, biased bridge is a better estimation algorithm since it takes into consideration of the training error and training relative biases. The results also show that biased bridge estimation achieved 100% accuracy on the SCUT-FBP5500 data and $92\%$ accuracy on the Adult Census Income data.**

## 6 CONCLUSION AND FUTURE WORK

In summary, this paper proposes relative fairness which checks whether one decision set is more biased towards a certain sensitive group than another decision set. Such relative fairness alleviates the need for defining what is considered to be absolutely fair. In addition, if there exists a reference set of decisions that is well-accepted to be fair in certain context, relative fairness of other decisions against that reference decision set can reflect their fairness in that context. In addition, two novel machine learning-based approaches are proposed to enable the testing of relative fairness between two decision sets made on different data. Assumptions and analyses are provided for when and how the proposed approaches work. Empirical results on a real world dataset with ratings from multiple humans and a dataset with synthetic biases showed that, the biased bridge approach achieved more accurate estimation than the unbiased bridge approach since it takes into consideration of the training error and training relative biases. This work suffer from several limitations:

**Limitation 1** According to **RQ1**, even decisions from the same humans may not always have the same trend of relative biases on different data. This suggests that the analysis of relative fairness requires a sufficient amount of decision data points.

**Limitation 2** According to **RQ2**, even the biased bridge estimation is not 100% accurate, there is still room for improvement.

**Limitation 3** The relative fairness and the two approaches are currently defined for binary sensitive attributes. New definitions are required for continuous sensitive attributes.

**Limitation 4** There is a risk of companies/humans using the relative fairness of their decisions against one specific reference decision set to justify their fairness in the decision making process. Without the definition of fairness for the context and external checking of the reference decision set, relative fairness can be misleading— a decision set can be biased when it is relatively fair against another biased decision set.

Overall, we believe that this work could benefit the community by presenting a new way of analyzing relative human biases with the help of machine learning models.

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
