# OpenReview forum: "Test Relative Fairness in Human Decisions With Machine Learning"
_ICLR.cc/2024/Conference — Submitted to ICLR 2024_

### Official Review · Reviewer_oWWd · 2023-10-22

**Soundness:** 1 poor
**Presentation:** 2 fair
**Contribution:** 2 fair
**Rating:** 3
**Confidence:** 3

**Summary:**

The paper presents a method to compute the relative fairness between two sets of decisions. The paper presents a couple of metrics to compute the relative fairness of the decisions. If one of the decision sets is incomplete the paper proposes to leverage a ML model to regress the ground truth decisions onto the unlabeled set. The paper presents two approaches to perform the same using an unbiased and a biased bridge. Empirical results show that the proposed metrics are consistent with the ground truth scores.

**Strengths:**

1. The paper is fairly easy to follow.
2. The notion of relative fairness is novel.

**Weaknesses:**

1. The premise and contribution of the paper is unclear to me. Specifically, the utility and application of the proposed metrics need more clarification. As I understand relative fairness is useful when comparing two decision sets. But if both decision sets are biased and the relative fairness is near zero, what does that tell us?
2. Relative fairness as presented in the paper is simply the difference between statistical parity (with some normalization) of each decision set. Why can't we just rely on the absolute statistical parity? It provides a more granular and absolute measure of fairness. Why do we need relative fairness or what extra information does that provide us?
3. It is unclear to me why ML models are being used to approximate the decisions on an unlabeled set. If a small set of human decisions is available, why can't we just evaluate the absolute fairness of that set? The function f can be prone to a range of implicit biases of its own and the results of the proposed metrics can heavily rely on the parameterization of f.
4. One of the use cases of relative fairness is when the fairness of the decisions suddenly changes over time. Even then it is unclear to me how relative fairness is useful over standard statistical parity.

**Questions:**

1. Table 3 and 4 are quite hard to read. What do the values in brackets indicate? Are those the p-values?
2. Please respond to the comments in the weakness section as well.

---

> ### Author Response · Authors · 2023-11-21
>
> We would like to thank the reviewer for pointing out the connection between the difference of statistical parity and our relative fairness definition. We have revised the paper accordingly to discuss the connection and difference between these two concepts. Please also find our detailed responses below:
>
> Question 1: Table 3 and 4 are quite hard to read.
> The values in brackets indicate the p values from the null hypothesis test of RBT. To make the tables easier to understand, we added some descriptions of the tables highlighted in Section 5 Page 7.
>
> Weakness 1: We have added Subsection 1.1 to show the potential applications of the proposed relative fairness definition and the estimation algorithms. We also rewrote Subsection 1.2 to clarify the contributions. We agree with the reviewer that it is possible to have two decision sets being relatively fair with each other when neither one is “absolutely” fair. In our opinion, this is perfectly fine and it does tell us that decision makers behind these two decision sets are consistent (if they are biased, they are biased in the same way). Because of the difficulty of specifying a standard for “absolute” fairness (as highlighted in Paragraph 2 Page 1), relative fairness is more general.
>
> Weakness 2: As highlighted in the last paragraph of Section 3.2 of the revised paper, we show that relative fairness is related to but not the same as the difference between statistical parity. Consider the following example:
>
> A   = [1,1,1,1,1]+[0,0,0,0,0,0]
>
> Y0 = [1,1,0,0,0]+[1,1,1,0,0,0]
>
> Y1 = [1,1,1,0,0]+[1,0,0,0,0,0]
>
> Y2 = [0,0,1,1,1]+[0,0,0,0,1,0]
>
> SP(Y0) = 2 / 5 - 3 / 6 = -0.1
>
> SP(Y1) = SP(Y2) = 3 / 5 - 1 / 6 = 0.43
>
> The difference between SP of Y1, Y0 is the same as the difference between SP of Y2, Y0 and is quite large -0.53.
> However, when tested for relative fairness, the p value for RBT(Y0, Y1) is 0.05 which means the differences between Y0 and Y1 are significantly different across A = 0 and A=1. But the p value for RBT(Y0, Y2) is 0.20 which means the differences between Y0 and Y2 are not significantly different across A = 0 and A=1. There is not enough evidence to draw the conclusion that Y0 and Y2 are relatively biased.
>
> We would prefer to call the relative fairness notion as difference parity since it focuses on analyzing the statistical parity of the decision differences (as added and highlighted in Section 3.1).
>
> Weakness 3: As we have shown in Weakness 2, the relative fairness definition is different from the difference between statistical parity. Therefore, we need to estimate the statistics of the differences before calculating for the relative bias metrics when decisions are made on different data. The concern of f being prone to a range of implicit biases of its own will affect the Unbiased Bridge algorithm but will be alleviated by the Biased Bridge algorithm. We have rewritten Section 3.3 to better describe how Biased Bridge alleviates such implicit biases. This is also why empirically, Biased Bridge performed better than Unbiased Bridge.
>
> Weakness 4: In terms of statistical parity vs relative fairness, we claim that
> (1) statistical parity is a notion for “absolute” fairness on one decision set; it requires the acceptance rates to be the same across different sensitive groups. When the actual acceptance rates are different across different sensitive groups, statistical parity can be misleading. On the other hand, relative fairness does not have any assumption on what is considered to be “absolutely” fair, thus can be more general.
> (2) the difference between statistical parity cannot reflect the statistical parity information of every decision difference like the relative fairness does. Thus we believe that in terms of evaluating relative fairness, our definition is better than just the difference between statistical parity.
>
> Again, we want to thank the reviewer for the detailed comments and suggestions. We hope that your questions and concerns have been properly addressed. Please let us know if there are further concerns.

---

### Official Review · Reviewer_XK1E · 2023-10-26

**Soundness:** 2 fair
**Presentation:** 2 fair
**Contribution:** 2 fair
**Rating:** 3
**Confidence:** 3

**Summary:**

The authors propose a different perspective of measuring fair decision-making by computing the fairness difference between decision sets made by various decision-makers. The authors evaluate the proposed approach on UCI-adult income and SCUT-FB500 face beauty datasets.

**Strengths:**

I like this perspective of measuring fair decision-making since there is potential to tap into often ignored aspects of decision-making: varied decision-maker biases and contextual differences.

**Weaknesses:**

While I liked the high-level idea, I think the problem modeling and presentation need some improvements. Below are the observed weaknesses (and respective suggestions) and some questions.

- The examples and general presentation of the paper are not easy to follow. For example, the authors mention various non-complementary examples, which doesn't help drive the idea home. The annotator bias example points more towards the setting of different (ideologically/demographically) decision-makers and the same dataset (e.g., platform content) or different datasets (e.g., different sub-reddit data/ different parts of X). On the other hand, the motivating example looks at multiple experts (>2, as is in the methodology) who are similar (ideologically) on the same dataset (similar features selected by a hiring company.). In general, the examples don't support or complement the methodology. Authors should pick an intuitive running example that supports the main idea.

- There is lots of inconsistency in writing. For example, the authors mention that they use different and same datasets in various parts of the write-up, which makes reading the paper hard. From my understanding, the authors use the same (exact) dataset and use part of the dataset as X_{0} (training set) and the other as X_{1} (validation set). Authors should aim for consistency in writing to improve readership.

- Although I liked the idea, it does look like theory and empirical designs don't support the goal. The definition of relative fairness would have been more robust and intuitive if the definition had constraints specific to decision-makers and data distributions. I believe varied constraint values significantly change the problem. The authors use historical datasets with no information on how/who collected the datasets, which is critical to the suggested framework.  Although individuals might be different by A/B, they might be ideologically/demographically or contextually (biasedly) similar, which changes the idea of the relative fairness subject to "different" decision-makers. Additionally, the authors use a beauty dataset which is somewhat not ethically a great decision, and mention that the adult dataset comes from truth and is, therefore, 100% fair, which is unrealistic and unverified.

  -  Authors should redefine (add constraints) relative fairness and respective terms and change datasets by using semi-synthetic datasets or conducting human subject studies to collect data. For example, authors could get content from different platforms or the same platform but different communities, and solicit for classifications from clearly different groups, e.g., 18-26 years and 27- 45 year-olds.
  -  Lastly, the methodology seems flawed to me, since the ML model and X_{0} biases are propagated and possibly amplified to decisions on X_{1}. Currently, it looks like authors view {X_{0}, Y_{0}} as ground truth for desigining f(x) which is then used on {X_{1}, Y_{1}}. I think instead, there should be a benchmark, for example, assuming you have information {X, Y} on prospective students, and you want to measure relative fairness. Then curate a distinct group A to classify X to get Y_{A}, and another group B to classify X to get Y_{B}. Then measure the relative fairness of the two decision sets.

- Although the authors cite some relevant literature, the literature review/related work section needs improvement. The section should be more cohesive and concise and clearly indicate the paper's strength and position in related works.

- The way authors mention their contributions is not complementary to the idea and their work. The authors should restructure this section and make it stronger. Additionally, the authors mention that they show that a machine-learning model trained on human decisions can inherit bias/preference. This is not novel and is the motivation for most of the fairness in ML work. For example, several works on feedback loops and bail/justice systems show this.

- Since authors define relative fairness as the statistical difference of the delta of two sets of decisions over a certain sensitive attribute, would it be reasonable to assume absolute fairness is particular to one decision set? Does it mean absolute fairness is a sub-routine of relative fairness?

**Questions:**

I like the idea but the paper presentation and methodology (theory and experiments) need some improvement. I have outlined some of these issues and made suggestions in the weakness section.

---

> ### Author Response · Authors · 2023-11-21
>
> First, we would like to thank the reviewer for the appreciation of our high-level idea. The reviewer also pointed out a direction of evaluating the relative fairness between decisions made for different tasks or in different contexts. While this is definitely a very interesting topic, we have to clarify that in this paper, we do not solve this problem. Given our current approach and experiments, we specified clearly in the revised paper (highlighted text on Page 2), this work only focuses on the relative fairness between different opinions (expert decisions) for the same task (dataset).
>
> Weakness 1: the examples and presentation of the paper.
> In the revised paper, we have replaced the previous motivating example with Subsection 1.1 Motivation. This subsection now clearly describes the example application scenarios of the proposed relative fairness evaluation and estimation algorithms.
>
> Weakness 2: inconsistency in writing.
> Since our previous writings about different data are confusing, we have made it clear in the revised version that, when we talk about different data, it refers to different data items from the same dataset (for the same task). So yes, X0 and X1 are always from the same dataset and are of the same distribution.
>
> Weakness 3: considering the demographics of the decision makers.
> Thanks for the suggestion! We totally agree with the reviewer that the bias or relative bias between decision makers can be related to their own demographics or ideologies. It would be great to experiment on a dataset containing the annotators’ demographic information. Unfortunately, we could not find such a dataset by far. We would much like to conduct such experiments in the future by collecting datasets with the annotators’ demographic information. However, this paper will not be able to include such experiments. In this paper, we focus on testing the relative bias/preference between two decision makers— e.g. Student A prefers Caucasian over Asian more than Student B does.
>
> We have clarified our usage of the Adult data in Section 4.1. Since it only has one set of labels, we synthesized different sets of labels with added Gaussian noise (favoring different sensitive groups). We also clarified in RQ1 of Section 5 that the purpose is to check whether the proposed relative fairness metrics are robust enough so that the relative fairness results on the training and test sets are consistent.
>
> As for the SCUT face beauty dataset, this is by far the only dataset we can find which includes complete decisions from different raters. This is the only real world dataset that can be used for our experiment. We would call the relative bias detected in this dataset relative preference. And such relative preference is no different from relative bias with one single difference— it does not raise any fairness concern. However, we still think the experiment on this data would be sufficient to answer our RQ1 and RQ2— whether the proposed relative bias metrics are consistent and whether the proposed estimation algorithms can correctly estimate the relative biases.
>
> Weakness 4: literature review.
> We have added some advantages of the proposed relative fairness over the existing ``absolute’’ fairness definitions in Section 2.
>
> Weakness 5: contributions.
> We have rewritten the contribution in Section 1.2 to highlight the novelty and contribution of the proposed relative fairness definition and the estimation algorithm. We have dropped the part of “a machine-learning model trained on human decisions can inherit bias/preference.”
>
> Weakness 6: absolute fairness as a sub-routine of relative fairness?
> We agree with the reviewer that (at least in this paper), absolute fairness is a concept considering one specific decision set. However, we do not completely understand what it means for absolute fairness being a sub-routine of relative fairness. We think relative fairness is about comparing two decision sets. The two decision sets can be relatively fair with each other even when either is not absolutely fair. Please clarify whether this answers your question.
>
> Again, we want to thank the reviewer for the detailed comments and suggestions. We definitely plan to explore the potential connection between the decision makers’ demographics and their relative biases in our future work. Please let us know if there are further concerns.

---

> > ### Comment · Reviewer_XK1E · 2023-11-22
> >
> > I highly appreciate the authors for addressing the raised questions and concerns and making several changes to the paper.
> > I am still convinced that the main idea is an interesting perspective with lots of potential and scale.
> > However, I do still have remaining issues, especially with the methodology.

---

> > > ### Author Response · Authors · 2023-11-22
> > >
> > > Thanks for your prompt feedback! Can you please specify what the remaining issues are with the methodology? Thank you!

---

> > > > ### Comment · Reviewer_XK1E · 2023-11-22
> > > >
> > > > As highlighted by all the reviewers (in the original reviews), there are several (unhandled) issues, even in edits, with the methodology and corresponding experiments, for example, the relative fairness metrics and the method's reliance on function f, trained on one of the decision sets (XO, YO), to estimate labels of the other decision sets (X1, Y1) is unclear.
> > > > Even with the few improvements made to the paper (methodology), the issues that could arise from the design are not effectively offset or efficiently handled.
> > > > In the current form, it is not clear what near-zero relative fairness signifies (take away is unclear) since decision-makers and their respective decision sets could be biased and not necessarily "absolutely" fair.

---

> > > > > ### Author Response · Authors · 2023-11-23
> > > > >
> > > > > Thanks for your feedback! We would like to clarify that
> > > > >
> > > > > 1. The unbiased bridge estimation does rely heavily on whether the model f(x) correctly learns the bias from the training data. However, it can be shown in the revised version that the biased bridge estimation does not rely on this and can accurately estimate the relative bias between Y0 and Y1 even when it’s predictions is not relatively fair against Y0 and Y1. We have adjusted in the Methodology section to make it clear that unbiased bridge only serves as a naive baseline.
> > > > >
> > > > > 2. As for the relative fairness itself, a zero relative bias suggests that these two decision makers are having similar biases/preferences. We believe that this information itself is useful in many scenarios especially when the standard for “absolute” fairness is ambiguous or hard to decide, e.g. when universities admitting students or when companies hiring employees.
> > > > >
> > > > > We hope this could clarify a few of your doubts. Thank you!

---

### Official Review · Reviewer_kf4f · 2023-10-31

**Soundness:** 1 poor
**Presentation:** 2 fair
**Contribution:** 1 poor
**Rating:** 3
**Confidence:** 4

**Summary:**

In fair classification, it is difficult or even ill-defined to have "true labels" for complex, subjective decision-making tasks. Hence, the paper proposes a relative notion of fairness. Operationally, for a pair of classifiers (people or algorithms), the paper tests for demographic parity (fraction of positives between two groups) between the two classifiers. The paper further considers a setting where the two classifiers make predictions (Y1 and Y2) on different datasets (X1 and X2). To apply the test, the paper trains a predictor on (X1, Y1), and use it to impute the first classifier's predictions on X2. The paper provides experimental results for the proposed methods on real datasets, in which the bias either exists in the data or is simulated.

**Strengths:**

1. In fairness research, it is important to think about the fairness notions carefully. The paper proposes a natural notion based on comparing a pair of classifiers in a relative sense, surpassing the need to define "true labels" which can be ambiguous or even infeasible in many applications.

2. The paper uses real data to evaluate the proposed method.

**Weaknesses:**

1. Methodology
Many data statistics have been used as the notion of fairness. Prior literature suggests that choosing which one to use requires careful thoughts. The paper should justify the choice of demographic parity, and ideally, provides comments on whether the proposed algorithms can generalize to other data statistics.

2. Novelty
Literature on individual fairness and label imputation should be overviewed:

(a) While the proposed relative comparison is natural, this idea of relative comparison is similar in spirit to individual fairness, in which "true labels" are also unnecessary.

(b) I appreciate simplicity in general, but I struggle to see the two proposed algorithms as "novel" as the paper has claimed. Essentially the paper trains a predictor to impute missing labels.

To these ends, I fail to see a clear main contribution of the paper: the formulation and ideas are not entirely novel; the theoretical result (Prop 4.5) is very weak; and the experimental results also need improvement (detailed in "Experiments").

3. Mathematical formulation
I find the mathematical formulation presentation quite loose:

(a) Def 4.1 is based on "equality in distribution". What is the source of randomness? I suppose it's the distribution of data X, but it is not formally stated that X is random.

(b) I find Assumption 4.2 quite problematic and unjustified. The assumption is on a complicated quantity (Y_\Delta(A=a)), and I fail to understand what this assumption fundamentally means. I especially struggle with Eq. 6, where normality is claimed without having any assumption on the function f or the classifiers (Y0 and Y1).

Is this the law of large numbers? Then the source of randomness needs to be defined. The assumption of large samples also in a certain sense defeats the purpose of label imputation in the "unbiased bridge" algorithm. One can just compute and compare the data statistics on (X1, Y1) and (X2, Y2) separately.

(c) I am confused about the correctness of Prop 4.5. In order to show equality in Def 4.1, under Assumption 4.2, should both equal mean and equal variance be required? Does RBT=RBD=0 only implies equal mean? Since the assumptions are very restrictive, this result somewhat feels like a tautology.

(d) Minor
- In Def 4.3, the mean and variance should be called "empirical mean" and "empirical variance".
- It should be made clear early in Sec 4 that Y is binary {0, 1}.
- In Eq. 9, what is \overline{f}?

4. Experiments
(a) I find that there is a non-trivial gap between the experimental setup, and the setting that the algorithms are introduced in. The problem is motivated by bypassing the need to define "true labels". However, the experiment still simulates settings that have these true labels. This also happens in Step 0 in in the motivating example of Section 3.

(b) The experiment setup needs to be described in more detail. For example, I don't understand two equations for the injection procedure. What does \overline{Y}_1 = \overline{Y}_0 mean? How is the mean of the Gaussian computed, since there are values of \lambda (separately for sex and race). What z(a) means also needs to be explained.

(c) Rigor: The paper needs to describe if multiple hypothesis testing is accounted.

(d) Presentation: The experimental results can be better presented by figures. The tables are very dense to parse.

5. Grammar:
- Title: "test" -> testing
- Abstract: "decisions sets"
- P2: "what are absolutely fair decisions" -> what absolutely fair decisions are
- P2: "present" -> presence
- P2: "Homan and others" -> refer to papers by the first author
- P7: "least beautify"

**Questions:**

Clarifications on Weaknesses 3(a, b, c), 4(a, b, c) would be the most helpful.

**Details Of Ethics Concerns:**

While I appreciate that the paper discusses limitations of the proposed approaches. I think the paper can make a more in-depth discussion. For example, what are the limitations of using the demographic parity metric, and what subpopulations may suffer more under this approach?

---

> ### Author Response · Authors · 2023-11-21
>
> First, we would like to thank the reviewer for the detailed comments. We have carefully considered all the comments and we believe that some of the comments have driven us to improve the paper greatly.
>
> Weakness 1. We have better justified the reason and choice of the proposed relative fairness notion in Paragraph 2 and 3 of Introduction (highlighted in the revision). In short, relative fairness can avoid the ambiguous and contradictory definitions of "absolute" fairness. Since we focus on the differences between two decision sets, it is natural to define it as the "statistical parity of the differences between two sets of decisions over a certain sensitive attribute." We also discussed the connection and difference between relative bias and demographic parity at the end of Page 4.
>
> Weakness 2a. We believe the proposed relative fairness is different from individual fairness in two aspects: (1) individual fairness evaluates one decision set (from human or algorithm) while relative fairness evaluates the relationship between two decision sets; (2) individual fairness believes that similar individuals (from different sensitive groups) should be treated similarly while relative fairness does not require that--- in scenarios where the sensitive attribute does have an impact on the outcome, relative fairness is still applicable but individual fairness is not.
>
> Weakness 2b. We agree that all the math in this paper is not new. However, we do believe this paper has a significant novelty and contribution--- (1) before this, there is no fairness definition based on the differences of two decision sets; (2) the biased bridge algorithm is not just label imputation. We have revised the contributions and the entire Methodology section to highlight these.
>
> Weakness 3ab. Inspired by the comments, we have dropped the assumption of normal distribution. The only assumption is that the differences of the two decision sets are i.i.d.. In this case, the sampled means of the differences should follow normal distribution based on the central limit theorem and the law of large numbers. The new definition is still different from just analyzing the statistics of the two decision sets individually DP(Y0)-DP(Y1). This is because it also takes into consideration the variance of the differences (Y0-Y1) and thus the biased bridge algorithm is still necessary. We also used an example to show the difference in our reply to Weakness 2 of Reviewer oWWd (the third reviewer).
>
> Weakness 3c. The unbiased bridge algorithm is very naive and is mainly used for comparison. We have dropped the analysis around it to focus more on the biased bridge algorithm.
>
> Weakness 3d. We have rewritten the Methodology section with those minor issues fixed. Note that in the definition, we clarify that Y0, Y1 $\in$ R and all the definitions and algorithms apply to not just binary classification but also regression problems (the SCUT face beauty experiment is also a regression problem).
>
> Weakness 4a. (1) we have revised the motivating example to Section 1.1 Motivations and provide two example scenarios. These new examples now discuss the potential use cases of the proposed relative fairness notion and algorithm when reference labels are not available. (2) In the Adult experiment, since only one set of labels is available, we treat it as a reference set and compare it against a synthetic label set. In the SCUT experiment, there are labels from different raters, we calculated the relative fairness between every pair. To evaluate the estimations of the biased bridge and unbiased bridge, we used the relative bias calculated with the actual ratings on both the training and test set Y1(X1)-Y0(X1), Y1(X0)-Y0(X0). In practice, Y0(X1) and Y1(X0) will be unavailable and the biased bridge algorithm will be applied to estimate the relative bias RBT(Y1,Y0) and RBD(Y1,Y0).
>
> Weakness 4b. We have revised the synthetic bias part in Section 4.1. We just used a straightforward way to add Gaussian noise to the original labels so that when $\lambda(A)>0$, a noise of positive mean is added to Y(A=1) and a noise of negative mean is added to Y(A=0). The two noises are of the same variance. z(a) and s(Y0) are used to set the injected bias onto the same scale (proportional to $\lambda(A)$) and to make sure that the overall mean of the injected bias is 0.
>
> Weakness 4c. We clarified in Section 5 that all experiments happened once.
>
> Weakness 4d. We failed to find a better way to present the results with figures--- mainly because we still want to present those numbers. However, we clarified the meaning of the colors in the tables in Section 5 Paragraph 1 so that it is possible for the readers to only look at the colors of the tables.
>
> Weakness 5. We have fixed all the grammar listed.
>
> Again, we want to thank the reviewer for the detailed comments and suggestions. Please let us know if there are further concerns.

---

### Author Response · Authors · 2023-11-22
**Global Author Rebuttal**

We would like to thank the reviewer for their detailed comments and suggestions. We have carefully considered all the comments and we believe that some of the comments have driven us to improve the paper greatly. The revised paper includes various clarifications highlighted in blue and more importantly, the Methodology section is completely rewritten to include more precise math following every reviewer's suggestions. We hope that the revised version has addressed all of the previous concerns, and we will also greatly appreciate the reviewers to continue to provide feedback on the revised version.

Thank you!

---

### Meta-Review · Area_Chair_CMHc · 2023-11-27

**Metareview:**

All the reviewers were negative about the paper and highlighted multiple areas for improvement, especially in regards of the methodology. The rebuttal period did not persuade them to increase/change their score.

**Justification For Why Not Higher Score:**

All the reviewers are negative about the paper.

**Justification For Why Not Lower Score:**

N/A

---

### Decision · Program_Chairs · 2024-01-16

Reject